# *Crocus sativus* L. Petal Extract Inhibits Inflammation and Osteoclastogenesis in RAW 264.7 Cell Model

**DOI:** 10.3390/pharmaceutics14061290

**Published:** 2022-06-17

**Authors:** Ciriana Orabona, Elena Orecchini, Claudia Volpi, Federico Bacaloni, Eleonora Panfili, Cinzia Pagano, Luana Perioli, Maria Laura Belladonna

**Affiliations:** 1Section of Pharmacology, Department of Medicine and Surgery, University of Perugia, Piazza Severi 1, 06129 Perugia, Italy; ciriana.orabona@unipg.it (C.O.); elena.orecchini@studenti.unipg.it (E.O.); claudia.volpi@unipg.it (C.V.); federico.bacaloni@studenti.unipg.it (F.B.); eleonora.panfili@unipg.it (E.P.); 2Department of Pharmaceutical Sciences, University of Perugia, Via del Liceo 1, 06123 Perugia, Italy; cinzia.pagano@unipg.it (C.P.); luana.perioli@unipg.it (L.P.)

**Keywords:** *Crocus sativus* L., macrophages, anti-inflammatory, osteoclastogenesis, bone disruption, circular economy

## Abstract

The dried stigmas of *Crocus sativus* L. (Iridaceae) are traditionally processed to produce saffron, a spice widely used as a food coloring and flavoring agent, which is important in the pharmaceutical and textile dye-producing industries. The labor-intensive by-hand harvesting and the use of only a small amount of each flower cause saffron to be the most expensive spice in the world. *Crocus* sp. petals are by-products of saffron production and represent an interesting raw material for the preparation of extracts intended for health protection in the perspective of a circular economy. In the present study, ethanolic extract from *Crocus sativus* L. petals (*Crocus sativus* L. petal extract, *Cs*PE) was tested on macrophages by in vitro models of inflammation and osteoclastogenesis. The extract was found to be endowed with anti-inflammatory activity, significantly reducing the nitric oxide production and IL-6 release by RAW 264.7 murine cells. Moreover, *Cs*PE demonstrated an anti-osteoclastogenic effect, as revealed by a complete inhibition of tartrate-resistant acid phosphatase (TRAP)-positive osteoclast formation and a decreased expression of key osteoclast-related genes. This study, which focuses on the macrophage as the target cell of the bioactive extract from *Crocus sativus* L. petals, suggests that the petal by-product of saffron processing can usefully be part of a circular economy network aimed at producing an extract that potentially prevents bone disruption.

## 1. Introduction

Every year high levels of waste and by-products are generated by the food industry. This is a considerable problem as these materials often hurt the environment and require considerable disposal costs. Recently, there is a growing interest in the re-evaluation of this biomass as it is considered a suitable source of bioactive molecules useful in the health field [1]. In this scenario, the saffron plant (*Crocus sativus* L.) is interesting and deserves to be investigated. Stigmas represent the noble and most used part of the plant. The main application of these is in the food industry as a spice; however, stigmas recently gained high attention in the health field due to the beneficial antioxidant properties of the molecules they contain, the most abundant of which are crocin, picrocrocin, safranal, and kaempferol [2,3]. The food chain’s processing of saffron leads to the production of by-products generally considered a waste (i.e., petals, tepals, spaths, corm, and tunics), which have been shown to contain molecules (generally phenolic compounds) with biological activity [4].

Petals are the most abundant by-product deriving from saffron harvesting. It is reported that, generally, 158,000–300,000 flowers are necessary to produce 1 kg of spice [5], with about 78 kg of petals discarded per 1 kg of stigmas [6,7]. Saffron petals are mainly used for animal feed or as natural dye for fibers [8,9]. However, it is noteworthy that saffron petals are rich in many antioxidant compounds [10], such as phenols, anthocyanins, and flavonoids [11]. Therefore, they represent an interesting raw material for the extraction of active molecules intended for use in the health field with different applications [12]. Phytopharmacological studies have demonstrated the efficacy of *Crocus sativus* L. petal as a protective agent against mild-to-moderate depression [13], hepatotoxicity [14], and hypertension [15]. In mice, the ethanolic and aqueous extracts of saffron petals have shown an antinociceptive effect against chemical-induced pain; the ethanolic extract was effective in reducing chronic, rather than acute inflammation [16].

Chronic inflammation occurs in several pathological states, including osteo-related diseases, such as osteoporosis, osteoarthritis, and other bone loss inflammatory conditions [17]. In the interaction between immune cells and bone cells, referred to as osteoimmunity, the macrophage–osteoclast axis plays a fundamental role [18].

Macrophages are mononuclear myeloid immune cells that typically engulf and digest microbes, dead cells, and debris by phagocytosis [19]. They play a crucial role against pathogen infection in both innate immunity, by releasing cytokines and nonspecific cytotoxic molecules, and adaptive immunity, by presenting foreign antigens to T cells for the subsequent specific reactivity of the immune system. In particular, proinflammatory macrophages of the M1 cell subset, activated by LPS and IFNγ, trigger inflammation-related responses by producing soluble factors (i.e., IL-6, IL-12, and TNF cytokines), and releasing nitric oxide (NO), a cytotoxic catabolite resulting from NO-synthase (NOS)-dependent degradation of arginine [20].

Osteoclasts are essential cells in the control of bone mass, which is continuously remodeled by cells that destroy (i.e., osteoclasts), and others that rebuild (i.e., osteoblasts), the bone matrix [21]. Responding to the receptor activator of the nuclear factor-kappa B (NF-κB) ligand (RANKL), a cytokine of the TNF family, and to M-CSF, a macrophage growth factor, macrophages can differentiate into osteoclasts by fusing one with the others to form multinucleated cells for increased resorption of large materials deriving from the bone disruption they will operate. In this differentiation process, osteoclasts become fully matured and completely activated after the induction of specific genes that mediate the osteoclasts’ bone remodeling function [22].

The macrophage–osteoclast axis is a critical component of the bone remodeling process, in which osteoblastogenesis and osteoclastogenesis are mutually regulated and in crosstalk with immune system cells. Dysregulation in this bi-directional crosstalk can lead to osteo-related inflammatory diseases, such as estrogen-deficient osteoporosis, autoimmune rheumatoid arthritis, and age-related osteoarthritis [17]. The macrophage–osteoclast axis is an interesting target for the treatment of bone damage, and, based on previous literature documenting the saffron petal extract’s ability to reduce chronic inflammation [16], we have deepened the existing poor research on this part of the *Crocus sativus* L. plant by investigating the anti-inflammatory activity and, for the first time, the possible anti-osteoclastogenic potential of saffron petals. Moreover, this study aims to propose an enhancement of saffron petals’ value, from a waste product of *Crocus sativus* L.’s stigma collection to a raw material for the extraction of the bioactive components.

Thus, in the present study, a saffron petal dry extract, obtained by a green and non-polluting method, was investigated by in vitro macrophage cell models for its possible ability to restrain the production of inflammatory mediators and contrast the osteoclast formation.

## 2. Materials and Methods

### 2.1. Chemicals and Reagents

Sigma Aldrich (St. Louis, MO, USA) purchased the reagents 3-(4,5-dimethylthiazol-2-yl)-2,5-diphenyltetrazolium bromide (MTT), LPS from *E. coli* serotype 055:B5, and Acid Phosphatase Kit 387-A, containing the tartrate-resistant acid phosphatase (TRAP) staining reagent and TRAP reaction buffer, which were prepared as described by the manufacturer’s instructions. Recombinant murine sRANKL (CHO) was obtained from Peprotech (PeproTech EC, Ltd., London, UK).

### 2.2. Plant Material, Extract Preparation, and Chemical Characterization

Saffron by-product (petals of *Crocus sativus* L., code 3802683 of C. V. Starr Virtual Herbarium), harvested in October 2019, was provided by the farm “UBI MAIOR” (Roccasalli, Accumoli, Italy). Immediately after the harvest, petals were frozen and freeze-dried to maintain their original features.

The extraction was performed following a procedure already used for other vegetal matrices [23,24] using ethanol/water 70/30 *v*/*v* as extraction solvent and a ratio of 2.46 g/200 mL (freeze-dried petals/solvent). This ratio was fixed after a preliminary evaluation of many petals/extraction solvent ratios to achieve the best yield of bioactive molecules by one extraction cycle. Freeze-dried petals were resuspended in the extraction solvent and macerated in dynamic conditions (magnetic stirring 1500 rpm) at 45 °C for 90 min. After this time, the supernatant was separated from the exhausted petals by vacuum filtration using filter paper, and then, the solvent was removed from the extract by rotary evaporator (Büchi, R-100) at the working temperature of 35 °C to preserve the original features of the extracted molecules. The concentrated product, solubilized in 25 mL of double-distilled water, was freeze-dried. The obtained dry extract, hereinafter called *Cs*PE, was stored at room temperature under CaCl_2_ until use. A preliminary qualitative evaluation of the extract’s phenolic composition was performed by ultra-performance liquid chromatography (UPLC). An UHPLC system Nexera XR (Shimadzu, Japan), equipped with an ACE Excel 2 C18-PFP column (10 cm × 2.1 mm) (ACE, Aberdeen, UK), was packed with particles of 2 μm. Flow rate during analysis was set to 300 μL/min, and the two mobile phases (phase A: 99% H_2_O, 1% CH_3_COOH; phase B: 100% acetonitrile) were eluted as follows: (i) linear increase in B phase from 5% to 30% in 3.0 min; (ii) linear increase in B phase from 30% to 40% in 1.4 min; (iii) linear increase in B phase from 40% to 60% in 2.5 min; (iv) linear increase in B phase from 60% to 99% in 1.5 min; and (v) isocratic in B phase at 99% for 0.5 min. Phenolic compounds of standard gallic acid, chlorogenic acid, epigallocatechin gallate (EGCG), quercetin 3-glucoside, quercetin, and kaempferol (Sigma Aldrich, Milan, Italy) were used as reference. Total duration of the analysis was of 12 min. Compounds’ identifications were performed through a QTRAP 4500 tandem mass spectrometer (Sciex, Concord, ON, Canada), coupled with an electrospray ionization source (V-source) operating in negative ionization mode as previously described [25].

### 2.3. Cell Cultures

The murine macrophage cell line RAW 264.7, obtained from the American Type Culture Collection (ATCC, Manassas, VA, USA), was cultured according to standard procedures in RPMI-1640 medium, supplemented with 10% heat-inactivated Fetal Bovine Serum (FBS), 2 mM of L-glutamine and antibiotics (100 U/mL penicillin, 100 μg/mL streptomycin). Cell cultures were used the day after vial thawing; they were never allowed to become confluent and were provided with a fresh medium every 3 days. Incubations were performed at 37 °C in a 5% CO_2_ atmosphere and humidified air. In the experiments, RAW 264.7 cells were seeded into wells of culture plates at a density of 5 × 10^4^ cells/cm^2^ and 1 × 10^4^ cells/cm^2^ for LPS- and RANKL-stimulated samples, respectively, and incubated overnight before treatment. The day after, supernatants were eliminated, and cells were treated with a medium containing scalar amounts of *Cs*PE (40 to 640 μg/mL) in the presence of LPS 50 ng/mL (proinflammatory model) or RANKL 100 ng/mL (osteoclastogenic model). After incubation, yellow-colored stimuli were eliminated by a cell wash prior to measuring cell viability and nitric oxide.

### 2.4. Cell Viability Assay

The cell viability of RAW 264.7 was measured by an MTT assay. Cells (5 × 10^4^ cells/cm^2^ and 1 × 10^4^ cells/cm^2^ for LPS- and RANKL-stimulated samples, respectively) were seeded into flat-bottom 96-well plates. After incubation with stimuli (*Cs*PE 40 to 5120 μg/mL) in the presence of LPS (50 ng/mL for 24 h) or RANKL (100 ng/mL for 5 days), adherent cells were washed once to remove any residual *Cs*PE dark yellow extract and incubated for 4 h at 37 °C with 110 μL of a medium containing MTT 50 μg. After the addition of 100 μL of solubilization buffer (SDS 10% in HCl 0.01M) to each well and an overnight incubation at 37 °C, the absorbance was read at 570 nm by a UV/visible spectrophotometer (TECAN, Thermo Fisher Scientific, Waltham, MA, USA). The assay was performed in triplicate for each concentration.

### 2.5. Nitrite Determination and IL-6 Measurement

RAW 264.7 cells (1.5 × 10^5^ cells/well) were allowed to attach to 96-well flat bottom plates overnight before the experiment. The day after, supernatants were eliminated, and cells were treated with a medium containing scalar concentrations of *Cs*PE (80 to 640 μg/mL) in the presence of LPS 50 ng/mL for 24 h. Then, RAW 264.7 adherent cells were washed to remove any residual *Cs*PE dark yellow extract, and incubation was resumed in a fresh culture medium for additional 24 h. To determine the total NO production, 50 μL of culture supernatant and 100 μL of Griess reagent (1 part of 1% naphthyl ethylenediamine dihydrochloride in distilled water plus 1 part of 1% sulfanilamide in 5% concentrate phosphoric acid) were allowed to react for 10 min, light-protected and at room temperature. The nitrite concentration was quantified via reading the sample absorbance at 540 nm by TECAN spectrophotometer (Thermo Fisher Scientific, Waltham, MA, USA) and comparing the obtained values to a sodium nitrite standard curve. The assay was performed in triplicate for each concentration. The mouse IL-6 uncoated ELISA Kit (Invitrogen, Carlsbad, CA, USA) was used to detect IL-6 production in culture supernatants as previously described [26] according to the manufacturer’s protocol (0.004 ng/mL, lower detection limit).

### 2.6. TRAP-Positive Cell Staining and Soluble TRAP Activity

TRAP-positive cells and soluble TRAP activity were assessed in RANKL-induced differentiation cultures by Acid Phosphatase Kit 387-A, (Sigma Aldrich, St. Louis, MO, USA) according to the manufacturer’s protocol. Briefly, 3 × 10^3^ RAW 264.7 cells per well (1×10^4^ cells/cm^2^) were allowed to adhere overnight to 96-well plates. The supernatant was then removed, and cells were stimulated for 5 days with different amounts of *Cs*PE (40 to 320 μg/mL) in the presence of RANKL 100 ng/mL to induce differentiation of RAW 264.7 macrophages into osteoclasts. The culture medium containing stimuli was changed every two days. After 5 days of incubation, TRAP activity was revealed at both the culture medium and cell levels. Thirty microliters of harvested supernatants were added to 170 μL of TRAP reaction buffer, and, after 2.5-h of incubation at 37 °C, absorbance at 540 nm was measured using a UV/visible spectrophotometer (TECAN, Thermo Fisher Scientific, Waltham, MA, USA). Adherent cells were fixed and stained by TRAP reagent at 37 °C for 1 h protected from light. Images of TRAP-positive cells were captured under a bright-field light microscope (EVOS M5000, Thermo Fisher Scientific, Waltham, MA, USA).

### 2.7. Real-Time PCR

After total RNA extraction by TRIzol (Invitrogen, Carlsbad, CA, USA) and reverse transcription to cDNA by QuantiTect Reverse Transcription Kit (Qiagen, Hilden, Germa-nia), a real-time PCR was performed using SYBR Green (Bio-Rad, Hercules, CA, USA) detection and the following specific primers: *Gapdh* forward 5′-CTGCCCAGAACATCATCCCT-3′; *Gapdh* reverse 5′-ACTTGGCAGGTTTCTCCAGG-3′; *Acp5* forward 5′-CTGCCTTGTCAAGAACTTGC-3′; *Acp5* reverse 5′-ACCTTTCGTTGATGTCGCAC-3′; *Calcr* forward 5′-TCATCATCCACCTGGTTGAG-3′; *Calcr* reverse 5′-CACAGCCATGACAATCAGAG-3′; *Mmp9* forward 5′-GCTGACTACGATAAGGACGGCA-3′; *Mmp9* reverse 5′-GCGGCCCTCAAAGATGAACGG-3′; *Ctsk* forward 5′-AGAAGACTCACCAGAAGCAG-3′; and *Ctsk* reverse 5′- CAGGTTATGGGCAGAGATTTG-3′ [27]. Values were calculated as the ratio of the specific gene to *Gapdh* expression, as determined by the relative quantification method (ΔΔCT; means ± SD of triplicate determination) [28].

### 2.8. Statistical Analysis and IC_50_ Calculation

All in vitro determinations are means ± SD from at least three independent experiments. Statistical significance was determined by the ANOVA one-way analysis (for different *Cs*PE concentrations, treated vs. untreated sample). The IC_50_ values were calculated by non-linear regression method [log(inhibitor) vs. response—variable slope] using GraphPad Prism Version 8.0.1 software.

## 3. Results

### 3.1. Characterization of CsPE Compounds

To determine its chemical composition, the ethanolic CsPE, prepared by the method described above, was subjected to chemical analysis by UHPLC. A preliminary qualitative evaluation of the CsPE’s composition revealed that it is characterized by a polyphenolic fraction with relevant amounts of gallic acid, chlorogenic acid, epigallocatechin gallate (EGCG), quercetin 3-glucoside, quercetin, and kaempferol, as shown in the obtained chromatograms (Figure 1).

### 3.2. Anti-Inflammatory Activity of CsPE on RAW 264.7 Macrophages

As macrophages display a pivotal role in driving innate immune responses, the well-established proinflammatory model of RAW 264.7 cells stimulated with LPS was used to investigate the putative anti-inflammatory activity of CsPE by quantifying the nitric oxide and IL-6 pro-inflammatory cytokine in the culture supernatant. Firstly, the viability of the cell system in response to CsPE was analyzed by incubating LPS-treated RAW 264.7 cells with serial two-fold dilutions of CsPE from 40 to 5120 µg/mL. After 24-h of exposure to the extract, cell viability was comparable to the LPS-activated control in the range of 40–640 µg/mL, maintained around 75% of the control at 1280 µg/mL, and strongly decreased at higher concentrations (Figure 2). Statistical analysis of LPS/CsPE-treated versus LPS-activated cells revealed significant (*p* = 0.0157) and highly significant (*p* < 0.0001) reductions in cell viability at 1280 μg/mL and at the two highest concentrations of CsPE (i.e., 2560 and 5120 μg/mL), respectively (Figure 2). The lowest five, not cytotoxic doses, ranging 80–640 μg/mL, were selected for assessing CsPE anti-inflammatory activity, evaluable as the inhibited release of pro-inflammatory NO and IL-6.

To investigate the anti-inflammatory effect of *Cs*PE on LPS-induced NO production, RAW 264.7 cells were co-treated with LPS 50 ng/mL and scalar amounts of *Cs*PE for 24 h. Nitrite accumulated in culture supernatants was dosed as an indicator of NOS2 activity and was significantly diminished by *Cs*PE in the range of 160–640 µg/mL, as compared to control cells treated with LPS alone (*p* = 0.0053 at 160 µg/mL, *p* < 0.0001 at the other two higher doses) (Figure 3a). The obtained concentration curves provided an IC_50_ = 257.8 ± 1.2 µg/mL (Figure 3b).

In the same cell system and at the same *Cs*PE concentration range, a potent inhibitory effect on the secretion of pro-inflammatory cytokine IL-6 was observed, too. Specifically, inhibition of the IL-6 secretion in the LPS-treated RAW 264.7 macrophages by *Cs*PE was highly significant already at the lowest extract concentration (80 µg/mL) (*p* < 0.0001), with IL-6 release dropping to one fifth compared to the LPS-control and becoming even more remarkable at higher *Cs*PE concentrations (Figure 4a). The extract’s inhibitory effect on IL-6 production was concentration-dependent with an IC_50_ = 24.8 ± 1.9 µg/mL (Figure 4b).

Overall, *Cs*PE exerted a potent anti-inflammatory effect by restraining NOS2 activity and reducing IL-6 release by LPS-exposed inflammatory RAW 264.7 macrophages in a viable range of concentrations from 80 to 640 µg/mL.

### 3.3. Inhibition of Osteoclastogenesis by CsPE

Pathological chronic inflammatory conditions are often accompanied by bone erosion, a recurring manifestation in arthritic states and aging [29]. To investigate whether CsPE might be endowed with bone-protective effects, in addition to anti-inflammatory activity, we turned to an in vitro model of osteoclast differentiation. Osteoclasts, derived from monocyte/macrophage lineage by cell fusion, are multinucleated cells that mainly contribute to bone erosion. Thus, RANKL-induced osteoclast differentiation from RAW 264.7 macrophages was used as a model to unveil the putative anti-osteoclastogenic potential of CsPE. Based on the strong cytotoxic effect observed in the LPS-activated RAW 264.7 cells in response to the highest concentration (5120 μg/mL) of CsPE (Figure 2), we exposed RANKL-differentiated RAW264.7 cultures to the range 40–1280 μg/mL for an MTT assay. No cytotoxic effect of CsPE was detected at 320 μg/mL and lower concentrations (Figure 5). On the contrary, a significant (*p* < 0.0001) reduction in cell viability, well below 75%, was observed at 640 and 1280 μg/mL. Based on these findings, the CsPE viable doses (40–320 μg/mL) were chosen to analyze the extract’s inhibitory activity on the osteoclast differentiation process.

Cultures of RANKL-differentiated and CsPE-treated RAW 264.7 cells were set up again, and, after 5-days, they were processed for TRAP staining, which is considered a marker of osteoclast function [30] (Figure 6a). The inhibitory effect of CsPE on RANKL-induced osteoclast differentiation from RAW 264.7 macrophages was evaluated by both counting the number of TRAP-positive purple–pink multinucleated cells having three or more nuclei and measuring the activity of the secreted soluble form of TRAP enzyme, selectively produced by differentiated osteoclasts in culture supernatants. Osteoclast formation was contrasted by CsPE treatment in a dose-dependent manner, so that TRAP-positive multinucleated cells were completely absent at 320 μg/mL (Figure 6a,b), with IC_50_ = 77.5 ± 1.3 µg/mL (Figure 6c). A significant reduction (*p* < 0.0001) of TRAP activity in the culture supernatant, attributable to the secreted soluble form of the enzyme, was also recorded for every tested concentration, being all the more relevant the greater the amount of CsPE (Figure 6d). Only at the highest dose of 320 μg/mL of CsPE, the absorbance of dark yellow extract at 540 nm wavelength was detectable in the absence of enzyme-producing cells (data not shown). Thus, a net absorbance value revealing the expected further decrease in TRAP activity at the highest dose is masked by the interference of CsPE absorbance at 540 nm.

The ability of CsPE to decrease RANKL-induced osteoclast formation was investigated at the gene level, too. For the TRAP encoding gene *Acp5*, a real-time PCR analysis confirmed significant (*p* = 0.0016 at 40 μg/mL), and highly significant (*p* < 0.0001 at higher doses) reductions in gene expression in CsPE-treated samples with respect to the untreated RANKL-differentiated control (Figure 7a). In particular, *Acp5* decreased in a dose-dependent manner, and its very low expression at 320 μg/mL mirrored the almost complete absence of TRAP-positive cells revealed by TRAP staining (Figure 6a). To further confirm the anti-osteoclastogenic effect of CsPE in this experimental model, we evaluated the expression of other markers of osteoclast formation and function, including calcitonin receptor (*Calcr*) (Figure 7b), matrix metallopeptidase 9 (*Mmp9*) (Figure 7c), and cathepsin K (*Ctsk*) (Figure 7d) [27]. *Calcr* gene, encoding for a protein involved in maintaining calcium homeostasis and in regulating osteoclast-mediated bone resorption, underwent a highly significant reduction (*p* < 0.0001) in the CsPE concentration range of 40–160 μg/mL and became undetectable at 320 μg/mL (Figure 7b). *Mmp9* (the gene for an endopeptidase degrading type IV and V collagens in the extracellular matrix), and *Ctsk* (encoding a lysosomal cysteine proteinase predominantly expressed in osteoclasts and involved in the bone remodeling and resorption), were both significantly downmodulated by *Cs*PE with respect to RANKL control, starting at 80 μg/mL (*p* = 0.0266 for *Mmp9*, and *p* = 0.0070 for *Ctsk*), and reaching very low (*Ctsk*) or almost undetectable (*Mmp9*) levels at 320 μg/mL of *Cs*PE (Figure 7c,d). Notably, for every tested gene, the expression was downregulated by *Cs*PE in a dose-dependent manner.

Therefore, CsPE can negatively modulate the RANKL-induced expression of the key genes responsible for osteoclast formation and function and effectively interfere with RANKL-dependent osteoclastogenesis, acting in a dose-dependent manner and exhibiting, among the tested concentrations, a complete inhibitory effect on osteoclast differentiation at 320 μg/mL.

## 4. Discussion

*Crocus sativus* is an herbaceous plant largely used in the food industry to derive saffron, a much-appreciated spice, from the stigmas. Recently, this noble part of the saffron flower has been also proved to be a source of bioactive, mainly antioxidant, ingredients (i.e., crocin, crocetin, picrocrocin, safranal) potentially exploitable in the health field [31]. Many studies have documented that crocin, which is extracted from saffron stigmas, exerts antioxidant and anti-inflammatory therapeutic effects in several diseases, including osteoarthritis [32] and osteoporosis, since it downregulates osteoclast differentiation via inhibition of JNK and NF-κB signaling pathways [33]. In particular, crocin revitalized cartilage and decreased bone deterioration along with inflammation and oxidative damage, and its administration prevented ovariectomy-induced osteoporosis in rats without hyperplastic effects on the uterus [34]. Moreover, both crocin and its hydrolyzed form crocetin effectively enhanced osteogenic differentiation of mesenchymal stem cells into osteoblasts and can be considered safe therapeutic agents in clinical applications of bone remodeling [35].

However, *Crocus sativus* L. stigmas, quite rare and very precious, generally are not intended to be used as a raw material for health products, both to avoid stealing biomass from the food sector and, mostly, because stigma-based formulations would be too expensive and inconvenient at the production and/or the final user level [36]. Nevertheless, saffron petals, which are considered the main waste deriving from stigma harvesting, are a valuable source of bioactive ingredients, as well. Several different extracts from *Crocus sativus* L. petals have shown protective effects in various experimental models. In a rat model of hypertension induced by angiotensin II and NG-nitro-L-arginine methyl ester (a NOS inhibitor), pre-treatment with a hydroalcoholic extract of saffron petals could significantly attenuate the cardiovascular responses [6]; the ethyl acetate fraction from *Crocus sativus* L. petals inhibited the nerve growth factor and brain-derived neurotrophic factor reuptake in rat C6 glioma cells model, accounting for the antidepressant properties of saffron petals [37]. In a 6-week double-blind, placebo-controlled and randomized trial of mild-to-moderate depression, patients that received capsules of *Crocus sativus* L. petals benefited from a significantly better outcome than those who received the placebo [13]; a hydroalcoholic extract of saffron petals displayed a protective effect against dysregulated ovarian steroids and inflammatory markers in a mouse model of testosterone-induced polycystic ovary syndrome [38]. Moreover, for saffron petal extracts, several studies documented the antioxidant and hepatoprotective activity in cell injury in vitro models [14,39,40,41], and the anti-inflammatory effect both in vitro [31] and in vivo [16]. Notably, although many of these studies use experimental models involving pro-inflammatory immune responses, only two works have thus far investigated macrophages as one of the possible cell targets of *Crocus sativus* L. petal derivatives, showing that two purified polysaccharides, not a whole petal extract, augmented the immunogenic functions of macrophage RAW 264.7 cells [42,43]. Thus, based on our recent research on the anti-arthritic activities of another plant extract [27], and pointing toward a possible application for saffron harvesting waste, the present study has investigated the ability of an ethanolic extract obtained from *Crocus sativus* L. petals in modulating specific macrophage cellular mechanisms involved in inflammation and osteoclastogenesis. Moreover, our study proposes the idea of integrating saffron harvesting waste (i.e., petals) into a circular economy system to be converted into a raw material for a phytopharmaceutical extract, useful in the control of inflammation and bone loss.

Among the biological effects of plant-derived extracts, modulation of immune response arouses great interest because inflammation is a key process in a variety of pathological conditions [44]. Thus, the control of inflammatory processes by natural compounds represents a topic of eminent importance for their potential application against several diseases, is directly linked to immune hyper-reactivity (autoimmune diseases, allergies, arthritis, etc.), but is also apparently unrelated to inflammation (e.g., hypertension, obesity, diabetes, cancer, etc.) [45,46,47].

The immune system has a relevant role in bone physiology and remodeling through the activity of specialized immune cells. Osteoblasts, derived from mesenchymal stem cells, are bone-forming cells that synthesize and deposit on the bone surface the proteins of the bone matrix and later undergo mineralization and calcification. Osteoclasts, which originate from hematopoietic stem cells and differentiate into monocytes and macrophages, are multinucleated bone-eating cells that become fully functional by RANKL stimulation [18,48]. This soluble ligand of RANK receptor is secreted, among other cells, by activated T lymphocytes and represents the most direct evidence of the immune system’s influence on bone resorption through osteoclasts [49]. Moreover, RANKL is positively regulated by proinflammatory cytokines IL-6, TNF-a, IL-1β, and IL-11, which are released by different cells in an inflammatory microenvironment and sustain further osteoclast formation and activity [50], leading to extensive bone damage in osteo-inflammatory diseases, such as osteoporosis, rheumatoid arthritis, and osteoarthritis [51]. Therefore, in bone-damaging diseases, since an overwhelming activation of osteoclasts with respect to the osteoblast compartment disrupts the physiological bone homeostasis, agents capable of restoring the lost balance could ensure the necessary therapeutic bone protection.

In our study, we found that *Cs*PE is effective in controlling both the critical mediators of inflammation and osteoclast differentiation. In the assessed macrophage cell line, *Cs*PE was much more effective in inhibiting LPS-dependent IL-6 production, than in controlling NO production, under the same in vitro cell conditioning. This activity could be attributed to gallic acid, the main compound found in *Cs*PE, which is a negative modulator of LPS-induced inflammatory responses in macrophages [52].

Differently from the inhibition of NO production (IC_50_ = 257.8 ± 1.2 µg/mL), the strong inhibitory effect on IL-6 secretion by *Cs*PE is quantified by an IC_50_ = 24.8 ± 1.9 µg/mL, which is a promising inhibitory activity for an herbal extract, comparable to that of small molecules in the early phase of drug discovery. Moreover, IL-6 is a key pro-inflammatory mediator in the pathogenesis of chronic bone consuming diseases, such as rheumatoid arthritis, and it plays the role of a drug target for the main anti-arthritic agents, such as monoclonal antibodies neutralizing either IL-6, or the specific receptor IL-6R [51], and small molecules interfering with IL-6 signaling (e.g., JAK inhibitors). Another therapeutic strategy against rheumatoid arthritis is the neutralization of RANKL by specific monoclonal antibodies, which inhibits bone erosion’s progression by sequestering the main signaling molecule for osteoclast differentiation [53]. This final effect is also obtained in our settings, where *Cs*PE demonstrated high efficacy in quenching the expression of genes critically involved in osteoclast formation and function. During OC polarization, the *Acp5* gene is induced, and the TRAP protein is expressed and actively secreted into the ruffled border and resorption lacuna (i.e., at the interface between the osteoclast plasma membrane and bone surface), where the TRAP’s enzymatic pro-form is cleaved by cathepsin K into the phosphatase-active protein degrading skeletal phosphoproteins, including osteopontin [54,55]. In the osteoclast’s functionally active cells, other upregulated genes are *Mmp9*, which encodes for a protein belonging to a family of extracellular matrix-degrading enzymes involved in tissue remodeling [56], and *Calcr*, which encodes for the calcitonin receptor, thereby maintaining calcium homeostasis and inhibiting osteoclast activity, and thus predisposing mature osteoclasts to be regulated in their bone resorption function [57]. The down-regulation of such bone disruption marker genes by *Cs*PE resulted in the repression of osteoclast differentiation with an IC_50_ = 77.5 ± 1.3 µg/mL, an intermediate value between those calculated for the inhibited production of NO and IL-6.

Studies reported by other authors showed that gallic acid and quercetin [58], as well as chlorogenic acid [59], can prevent osteoclast differentiation and bone loss, which suggests that these two molecules, found in *Cs*PE, could be the main molecules responsible for the observed effect. It could be speculated that, in the *Cs*PE phytocomplex, these two polyphenols might work synergistically to produce more effective benefits against bone loss than those obtained by the single molecules, a hypothesis worthy of confirmation by future studies.

The revaluation of saffron petals has been performed by eco-friendly, simple, scalable, and cheap procedures, guided by the criterion of sustainable development. Thus, petals were freeze-dried after harvesting to maintain their original features, a “clean” procedure largely used in the food industry. For the extraction, a hydroalcoholic solution, containing non-polluting organic solvents, was used. The solvent, enriched with extracted active ingredients, was recovered by distillation to be further reused. The exhausted petals, collected after filtration, have been made available for other applications in different fields (e.g., in the packaging industry). Notably, all of the steps requiring mechanical machinery and equipment took place at low temperatures and, in a short time, allow for very low consumption of energy and water as well as reduce the production of carbon dioxide, so that the sustainability of the whole process was improved.

Although the plant world has always provided interesting and useful remedies for human health, this sector is sometimes not highly exploited and is often overwhelmed by drug synthesis. Nowadays, plants still represent significant sources of pharmacologically active compounds and provide considerable pools of molecules for identifying novel drug leads. Nevertheless, a strong point of phytopharmacology is that, in many cases, the plant’s crude extract is endowed with a higher efficacy compared to its isolated components. This effect occurs because adjuvant substances, present in the whole extracts, synergistically enhance the active principles’ efficacy, protecting them from enzyme-mediated degradation, facilitating their transport across cell barriers, helping them to overcome multi-drug resistance mechanisms, or providing additional effector signals [60]. To understand whether *Cs*PE is more effective in inhibiting osteoclast formation than its isolated compounds, we plan in the near future to study selected molecules among those that will be identified as constituents of this saffron petal extract.

## 5. Conclusions

In the present work, for the first time, we demonstrate that a saffron petal extract can effectively inhibit osteoclast differentiation in vitro, thereby unveiling that *Crocus sativus* L. petals can be a resource in the control of inflammation and bone disruption occurring in osteo-related diseases. The extract was obtained following the principles of “green technology”, which is an important criterion in the transition from a linear to circular economy model, aiming to greatly reduce waste and the environmental impact of industrial production. Thus, this saffron petal extract represents an example of the eco-friendly transformation of waste into a valuable health product.

## Figures and Tables

**Figure 1 pharmaceutics-14-01290-f001:**
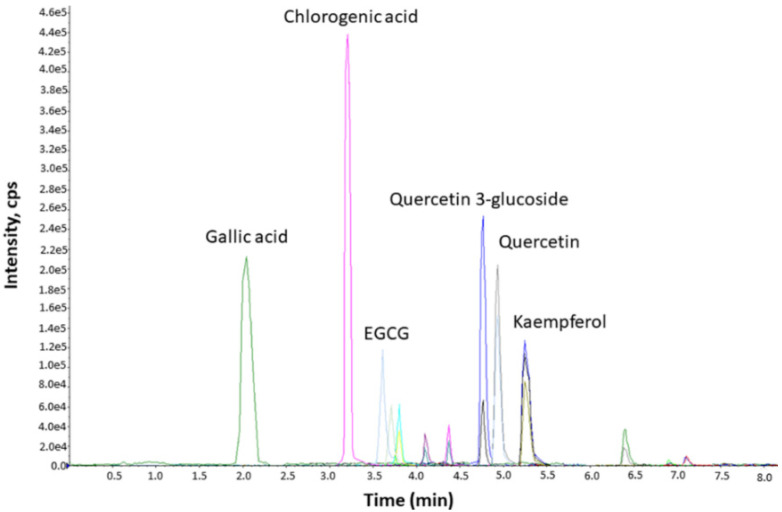
Chemical composition of CsPE. The ethanolic extract was analyzed by UHPLC. Chromatograms of the main compounds identified in CsPE were obtained by ion-trap tandem mass spectrometry (UHPLC-QATRAP) using an ACE Excel 2 C18-PFP column.

**Figure 2 pharmaceutics-14-01290-f002:**
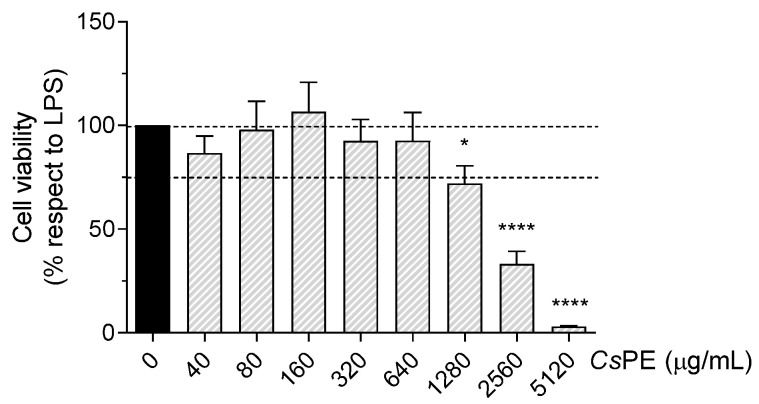
Cell viability analyzed by MTT assay on LPS-activated RAW 264.7 treated with *Cs*PE. RAW 264.7 cells were co-treated for 24 h with LPS 50 ng/mL and *Cs*PE at the indicated concentrations. The percentage of viable cells with respect to LPS-activated control was evaluated by MTT assay and reported as the mean ± SD of three independent experiments, each conducted in triplicate. Dotted lines indicate 100% and 75% of control viability. * *p* < 0.05, **** *p* < 0.0001, LPS/*Cs*PE-treated (grey bars) vs. LPS-activated cells (black bar) (one-way ANOVA test).

**Figure 3 pharmaceutics-14-01290-f003:**
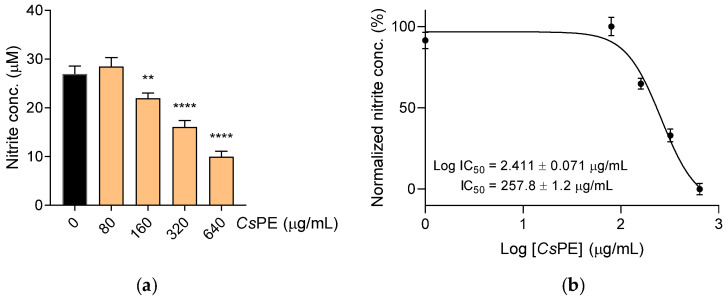
Inhibition of NO release in proinflammatory cultures of LPS-activated RAW 264.7. RAW 264.7 cells were cultured with LPS alone (control, black bar) or in combination with different amounts of *Cs*PE for 24 h (orange bars). (**a**) NO released in the culture supernatant was quantified as nitrite concentration by using Griess reagent. Results are reported as mean ± SD of three independent experiments, each conducted in triplicate. ** *p* < 0.01, **** *p* < 0.0001, LPS/*Cs*PE-treated versus LPS-activated cells (one-way ANOVA test). (**b**) A concentration-response curve was obtained to determine the IC_50_ of *Cs*PE. The IC_50_ value was determined using non-linear regression analysis.

**Figure 4 pharmaceutics-14-01290-f004:**
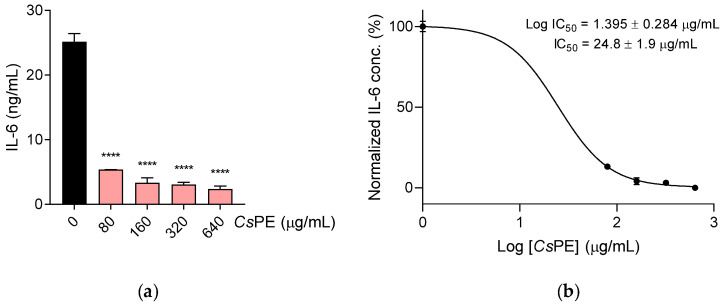
Inhibition of IL-6 secretion in proinflammatory cultures of LPS-activated RAW 264.7. RAW 264.7 cells were cultured with LPS alone (control, black bar) or in combination with different amounts of *Cs*PE for 24 h (pink bars). (**a**) IL-6 concentration in the culture supernatants was determined by the ELISA test. Results are reported as the mean ± SD of three independent experiments, each conducted in triplicate. **** *p* < 0.0001, LPS/*Cs*PE-treated versus LPS-activated cells (one-way ANOVA test). (**b**) A concentration-response curve was obtained to determine the IC_50_ of *Cs*PE. The IC_50_ value was determined using non-linear regression analysis.

**Figure 5 pharmaceutics-14-01290-f005:**
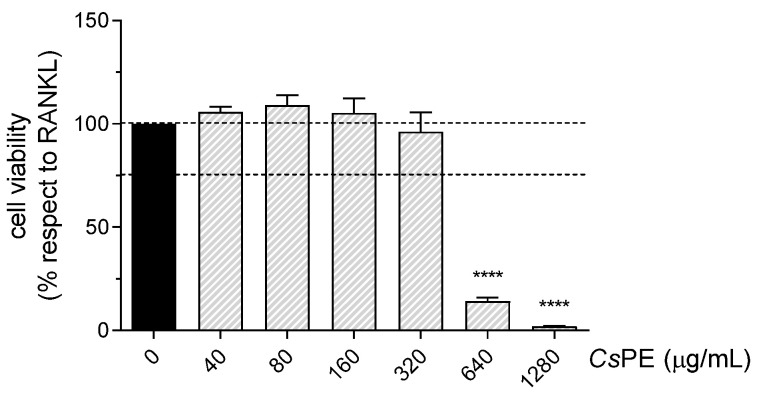
Cell viability analyzed by MTT assay on RANKL-differentiated RAW 264.7 treated with *Cs*PE. RAW264.7 cells were co-treated for 5 days with RANKL 100 ng/mL and *Cs*PE at the indicated concentrations. The percentage of viable cells with respect to RANKL-differentiated control was evaluated by MTT assay and reported as the mean ± SD of three independent experiments, each conducted in triplicate. Dotted lines indicate 100% and 75% of control viability. **** *p* < 0.0001, RANKL/*Cs*PE-treated (grey bars) versus RANKL-differentiated cells (black bars) (one-way ANOVA test).

**Figure 6 pharmaceutics-14-01290-f006:**
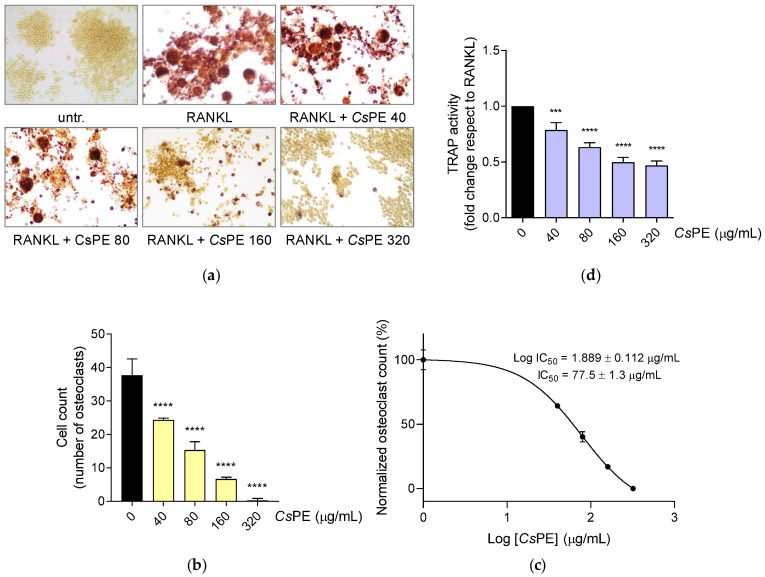
Suppression of osteoclast differentiation by CsPE in RAW 264.7 cultured with RANKL. (**a**) Representative staining of TRAP-positive osteoclasts, induced by RANKL from RAW 264.7 macrophages in the presence of the indicated concentrations of CsPE; untr, RAW 264.7 cells cultured without RANKL and left untreated. Osteoclasts are TRAP-positive purple–pink multinucleated cells having three or more nuclei. (**b**) Quantitative analysis of osteoclasts represented as TRAP-positive multinucleated cell number in RANKL-differentiated control (black bar) and CsPE/RANKL-treated samples (yellow bars). Results are reported as mean ± SD of the osteoclasts’ number counted in randomly selected visual fields in different areas of each well. Only TRAP-positive purple–pink cells with three or more nuclei were considered osteoclasts. (**c**) A concentration-response curve was obtained to determine the IC_50_ of CsPE. The IC_50_ value was determined using a non-linear regression analysis. (**d**) Quantitative analysis of TRAP activity in the culture supernatant, represented as fold change in CsPE/RANKL-treated samples (violet bars) with respect to RAW 264.7/RANKL control (black bar) and reported as the mean ± SD of three independent experiments, each conducted in triplicate. *** *p* < 0.001, **** *p* < 0.0001, CsPE/RANKL-treated versus RANKL-differentiated control (one-way ANOVA test).

**Figure 7 pharmaceutics-14-01290-f007:**
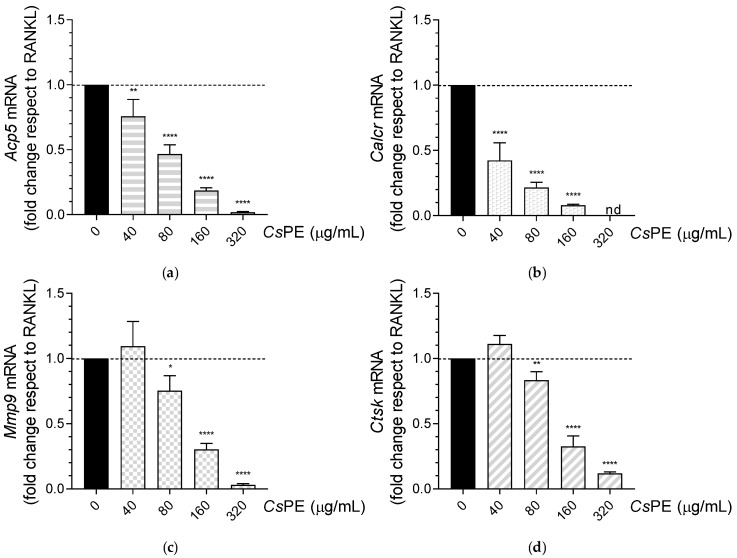
Down-regulation of marker genes of osteoclast formation and function in RANKL-differentiated RAW 264.7 cells treated with *Cs*PE. Real-time PCR analysis of (**a**) *Acp5*, (**b**) *Calcr*, (**c**) *Mmp9*, (**d**) and *Ctsk* transcripts in RAW 264.7 differentiated with RANKL and treated with the indicated concentrations of *Cs*PE. Gene expressions were normalized to the expression of *Gapdh* and reported as relative to the normalized expression in RANKL-differentiated control (black bar; dotted line, fold = 1); nd, not detectable. Data (mean ± SD) are the results of three independent measurements. * *p* < 0.05, ** *p* < 0.01, **** *p* < 0.0001, *Cs*PE/RANKL-treated versus RANKL-differentiated control (one-way ANOVA test).

## Data Availability

Not applicable.

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
