# Peer review of "Crocus sativus L. Petal Extract Inhibits Inflammation and Osteoclastogenesis in RAW 264.7 Cell Model"

_pharmaceutics, 2022, doi:10.3390/pharmaceutics14061290_

Round 1
Reviewer 1 Report
Congratulations to the authors. This is an interesting paper, the study is relevant, and the author contribute to this field of research.
The title reflects the main purpose of the manuscript precisely. The methods used and the results were well described, the conclusions are relevant.
I have only a few comments: The authors should use it correctly the Latin name of the plant in the whole article, respectively Crocus sativus L. Of the studied species a specimen must be kept in a recognized herbarium. Please specify this herbarium and registration code. The authors need to check the list of references and correct it according to the Instructions for authors.
Author Response
We would like to thank Reviewer 1 for the punctual comments and suggestions that contributed to ameliorating the quality of the manuscript.
We have made the changes suggested by Reviewer 1 (the correct Latin name Crocus sativus L. now has been used, and the herbarium registration code now has been added; moreover, the whole list of References has been updated according to the MDPI Full Reference Formatting Guide).
Best regards,
Maria Laura Belladonna
Reviewer 2 Report
The paper is well written but needs some revising before being published.
“Anti-osteoclastogenic activity of Crocus sativus petal extract: 2 turning saffron waste into a valuable resource against bone disruption”
The title reads more like a review article. Changing to title to be more descriptive of experimental would be beneficial. At the same time, such a broad statement is misleading because these experiments were done only in-vitro. I would suggest “Crocus sativus petal extract inhibits inflammation and osteoclastogenesis in RAW 264.7 cell model”
The abstract reads well.
The authors should expand on the work already done with saffron petals. Was any work done with macrophages?
(i.e., crocin, the most abundant, picrocrocin, safranal, kaempferol) is hard to understand. What is the most abundant and what of?
How many kg of petals are discarded per pound of saffron?
“Osteoclasts are essential cells in the control of bone mass, continuously destroyed by 70 osteoclasts and reformed by osteoblasts [18].” Clarify this sentence.
Why was 2.46g/200ml chosen? The number seems very arbitrary.
More methodology needs to be written for UPLC. Is the method identical to previously described? Was it modified in any way?
More detail is needed on the method and software that was used to determine the IC50.
Was cell viability tested with the extract and LPS at the same time?
The primer sequences should be published with this manuscript. Did the authors design new primers or obtained them in literature?
The compound of interest quantity should be calculated. The compounds are not exotic and should have standards.
Author Response
We would like to thank the Reviewer 2 for the punctual comments and suggestions that contributed to ameliorating the quality of the manuscript. We have answered point by point , as follows:
- “Anti-osteoclastogenic activity of Crocus sativus petal extract: 2 turning saffron waste into a valuable resource against bone disruption” The title reads more like a review article. Changing to title to be more descriptive of experimental would be beneficial. At the same time, such a broad statement is misleading because these experiments were done only in-vitro. I would suggest “Crocus sativus petal extract inhibits inflammation and osteoclastogenesis in RAW 264.7 cell model”
The title has been changed as suggested by the Reviewer.
- The authors should expand on the work already done with saffron petals. Was any work done with macrophages?
Lines 405-427 (518-540 in the “Track Changes” format): studies on different Crocus sativus L. petal extracts have been reported, including the two works that have so far investigated the effect of saffron petal derivatives (in those studies, two polysaccharides) on macrophages.
- (i.e., crocin, the most abundant, picrocrocin, safranal, kaempferol) is hard to understand. What is the most abundant and what of?
Line 41 (41 in the “Track Changes” format): the sentence has been modified into ‘The main application of them is in the food industry as a spice, however, stigmas recently gained high attention in the health field due to the mainly antioxidant beneficial properties of the molecules they contain, the most abundant of which are crocin, picrocrocin, safranal, and kaempferol’.
- How many kg of petals are discarded per pound of saffron?
Lines 48 (51 in the “Track Changes” format): information about Kg of petals discarded per Kg of saffron has been included, as required.
- “Osteoclasts are essential cells in the control of bone mass, continuously destroyed by 70 osteoclasts and reformed by osteoblasts [18].” Clarify this sentence.
Lines 71-73 (74-76 in the “Track Changes” format): the sentence has been modified into ‘Osteoclasts are essential cells in the control of bone mass, which is continuously remodeled by cells that destroy (i.e., osteoclasts) and others that rebuild (i.e., osteoblasts) the bone matrix’.
- Why was 2.46g/200ml chosen? The number seems very arbitrary.
Lines 112-114 (117-119 in the “Track Changes” format): The ratio was fixed after a preliminary evaluation of many petals/solvent ratios. The suitable conditions were obtained using 2.46 g/200 ml freeze-dried petals/extraction solvent as we observed that in these conditions the best yield of bioactive molecules could be obtained by one extraction cycle.
- More methodology needs to be written for UPLC. Is the method identical to previously described? Was it modified in any way?
Lines 122-134 (127-139 in the “Track Changes” format): The UPLC detailed methodology has been included in the M&M section, as requested.
- More detail is needed on the method and software that was used to determine the IC50.
Lines 207 and 210-212 (319 and 322-324 in the “Track Changes” format): method and software used to determine the IC50 have been included in the M&M section, as requested.
- Was cell viability tested with the extract and LPS at the same time?
Lines 154-155 (170-171 in the “Track Changes” format): cell viability was tested ‘After incubation with stimuli (CsPE 40 to 5120 μg/ml) in the presence of LPS (50 ng/ml for 24 hours)’, as reported in the M&M section.
- The primer sequences should be published with this manuscript. Did the authors design new primers or obtained them in literature?
Lines 197-204 (309-316 in the “Track Changes” format): sequences of already used primers now have been reported, as requested.
- The compound of interest quantity should be calculated. The compounds are not exotic and should have standards.
Lines 129-131 (134-136 in the “Track Changes” format): the standards used for the analysis have been included in the method section, as required.
Best regards,
Maria Laura Belladonna
Round 2
Reviewer 2 Report
The authors' revision sufficiently improved the manuscript for publication.